# Taxonomy of Acute Stroke: Imaging, Processing, and Treatment

**DOI:** 10.3390/diagnostics14101057

**Published:** 2024-05-19

**Authors:** Wieslaw L. Nowinski

**Affiliations:** Sano Centre for Computational Personalised Medicine, Czarnowiejska 36, 30-054 Krakow, Poland; w.nowinski@sanoscience.org

**Keywords:** stroke, acute stroke, stroke management, stroke imaging, neuroimage processing, artificial intelligence, brain atlases, stroke treatment, thrombolysis, thrombectomy

## Abstract

Stroke management employs a variety of diagnostic imaging modalities, image processing and analysis methods, and treatment procedures. This work categorizes methods for stroke imaging, image processing and analysis, and treatment, and provides their taxonomies illustrated by a state-of-the-art review. Imaging plays a critical role in stroke management, and the most frequently employed modalities are computed tomography (CT) and magnetic resonance (MR). CT includes unenhanced non-contrast CT as the first-line diagnosis, CT angiography, and CT perfusion. MR is the most complete method to examine stroke patients. MR angiography is useful to evaluate the severity of artery stenosis, vascular occlusion, and collateral flow. Diffusion-weighted imaging is the gold standard for evaluating ischemia. MR perfusion-weighted imaging assesses the penumbra. The stroke image processing methods are divided into non-atlas/template-based and atlas/template-based. The non-atlas/template-based methods are subdivided into intensity and contrast transformations, local segmentation-related, anatomy-guided, global density-guided, and artificial intelligence/deep learning-based. The atlas/template-based methods are subdivided into intensity templates and atlases with three atlas types: anatomy atlases, vascular atlases, and lesion-derived atlases. The treatment procedures for arterial and venous strokes include intravenous and intraarterial thrombolysis and mechanical thrombectomy. This work captures the state-of-the-art in stroke management summarized in the form of comprehensive and straightforward taxonomy diagrams. All three introduced taxonomies in diagnostic imaging, image processing and analysis, and treatment are widely illustrated and compared against other state-of-the-art classifications.

## 1. Introduction

A stroke, or brain attack, is an abrupt onset of neurological injury resulting from disease (occlusion or rupture) of the arteries or veins serving the central nervous system [1]. Stroke often results in the impairment of movement, sensation, and/or function. Cerebrovascular disease can progress from diverse causes, comprising thrombosis, atherosclerosis, cerebral venous thrombosis, or an embolic arterial blood clot. Stroke is the most common life-threatening neurological disorder, a leading cause of death, and a main cause of permanent disability. It has a deep impact on public health and generates vast costs for primary treatment, hospitalization, rehabilitation, and chronic care. Stroke management is demanding because of the variety of diagnostic imaging modalities, image processing and analysis methods, and treatment criteria and protocols as well as time restrictions to rapidly evaluate, make a decision, and treat, among others [2,3,4,5,6,7].

Imaging plays a vital role in stroke management, as addressed in several textbooks and papers [3,8,9,10,11]. In order to make the diagnosis and therapeutic decision, multiple images usually need to be acquired. Stroke appearance on neuroimages varies with infarct age. The temporal evolution of strokes is usually classified into hyperacute (0–6 h), acute (6–24 h), subacute (24 h to approximately 2 weeks), and chronic (more than 2 weeks old) [8]. The key questions in the imaging evaluation of stroke are: Is there hemorrhage? Is there an intravascular thrombus causing the vessel occlusion that can be removed? Is there an infarct, i.e., tissue that is irreversibly damaged and non-salvageable? Is there a penumbra, i.e., tissue at risk of progressing to infarction in the absence of rapid reperfusion that is still potentially salvageable if reperfused? The most frequently employed imaging modalities are computed tomography (CT) and magnetic resonance (MR) imaging. CT sensitivity to ischemia is only 25% versus 86% in MR. In addition, within the first 3 h, this sensitivity is lowered to 7% for CT and 46% for MR [12].

Numerous neuroimage processing and analysis methods have been developed for stroke management based on various criteria and techniques, as reviewed in several papers covering CT [13,14,15], MR and CT [16,17], and artificial intelligence (AI) and deep learning (DL) techniques [18,19,20,21,22,23]. They facilitate the interpretation of stroke scans and assist in decision-making in stroke management. Fast processing of stroke neuroimages is particularly crucial, as the time window to treat ischemic stroke by standard intravenous thrombolysis is only 4.5 h from stroke onset to therapy [24,25]. Stroke therapy can be pharmacologic, surgical, and interventional, and this subject is covered in several textbooks and review papers [3,10,26,27,28].

The goal of this work is to capture the state-of-the-art in stroke management in three areas, namely, stroke imaging, image processing and analysis, and treatment. The methods in these areas are featured, widely illustrated, and categorized and their corresponding taxonomies are summarized in the form of comprehensive and straightforward diagrams. In addition, the introduced taxonomies are compared against other classifications.

## 2. Taxonomy of Stroke Imaging

Acute stroke is a medical emergency and diagnostic imaging must be promptly and accurately performed and quickly interpreted so as not to delay treatment. Neuroimaging plays a crucial role in stroke management to (1) differentiate stroke from non-strokes and stroke-mimicking conditions, such as a brain tumor, brain abscess, toxic and metabolic disturbances, seizures, migraine, meningitis, or encephalitis; (2) differentiate between an ischemic and hemorrhagic stroke; (3) identify or exclude vessel occlusion, and if present, provide its site; (4) evaluate the collateral vascular network; (5) identify in acute ischemic stroke the infarct (core) along with its site and size; (6) identify in acute ischemic stroke the penumbra along with its site and size; (7) identify any chronic infarct(s); and (8) identify stroke subtypes. The two main groups of stroke diagnostic imaging modalities are CT and MR. A diagram illustrating the taxonomy of stroke imaging is shown in Figure 1.

CT includes three modalities useful for stroke: unenhanced non-contrast CT (NCCT), CT angiography (CTA), and CT perfusion (CTP). NCCT is the first-line diagnosis for emergency evaluation of acute stroke [29] due to its widespread availability, speed, and low cost. It allows for the exclusion of hemorrhage, which is the main contraindication for revascularization treatments. It identifies 90–95% of subarachnoid hemorrhages and almost 100% of intraparenchymal hemorrhages and assists in ruling out non-vascular causes for neurological symptoms [26]. Therefore, NCCT value is mostly in the initial evaluation and the exclusion of hemorrhagic stroke patients from thrombolytic treatment, whereas CTA and CTP significance is in selecting these patients who can benefit from thrombolysis. CTA provides imaging of the cervical and cerebral arteries and identifies or excludes large-vessel occlusion, detects stenosis, and evaluates the collateral vascular network, particularly with multiphase CTA. Therapeutically, it is also recommended for endovascular reperfusion. CTP, by imaging of the inflow and outflow dynamics of the contrast agent in the parenchyma, allows for the calculation of perfusion maps useful for the evaluation of the infarct core and the penumbra zone. In practice, NCCT, CTA, and CTP are combined (known as “CT+”) to enable a decision concerning stroke therapy.

MR is the most complete method to examine stroke patients. It has high sensitivity and specificity for the diagnosis of ischemia in the early hours after stroke onset [26]. Multiple conventional and advanced MR studies demonstrate normal and pathological anatomy, angiography, diffusion, perfusion, and spectroscopy by employing numerous pulse sequences [30]. Anatomy and certain parenchymal abnormalities are revealed with structural imaging, such as T1-weighted (T1W), T2-weighted (T2W), T2* gradient echo (GRE), fluid-attenuated inversion recovery (FLAIR), and susceptibility-weighted imaging (SWI). T2* enables the exclusion of hemorrhage, including small focal parenchymal hemorrhages, and SWI reveals microangiopathy, whereas T2 and FLAIR detect 90% of acute infarcts by 24 h [8]. MR angiography (MRA) is valuable for evaluating the severity of artery stenosis, vascular occlusion, and collateral flow. Diffusion-weighted imaging (DWI) is the gold standard for evaluating ischemia. It is highly sensitive and specific in the detection of acute ischemic infarcts earlier than CT and conventional MR. DWI identifies the ischemic area as early as 35 min after the onset of symptoms [26]. Moreover, DWI along with the apparent diffusion coefficient (ADC) calculated from DWI quantifies the degree of diffusion that allows the detection and assessment of the infarct and distinguishing new from chronic infarcts. MR perfusion-weighted imaging (PWI) examines hemodynamic conditions at the microvascular level, enabling the evaluation of the penumbra. Perfusion maps with various hemodynamic parameters can be computed from the CTP or PWI, and they include cerebral blood flow (CBF), cerebral blood volume (CBV), mean transit time (MTT), which is calculated as CBV/CBF, time to peak (TTP), and time to maximum (Tmax). MR spectroscopy images intracellular metabolites and can serve as a surrogate for stroke treatment.

Other stroke imaging modalities include ultrasonography (US), digital subtraction angiography (DSA), positron emission tomography (PET), and single-photon emission computed tomography (SPECT).

US, including Doppler and color duplex sonography, is a noninvasive procedure that allows for the evaluation of vascular pathologies. Color duplex sonography can provide hemodynamic information, such as stenosis, occlusion, and collaterals, as well as morphological findings, including plaques and hypoplasia. Transcranial Doppler can be employed for diagnosis of major cerebral artery occlusions and monitoring the effects of thrombolytic therapy [31]. In addition, US applied in ultrasonic thrombectomy treatment potentially increases the recanalization rate [10].

DSA is a catheter angiography that allows for the evaluation of vascular pathologies, although it is less commonly used diagnostically because of its invasiveness and lower sensitivity and specificity compared to CTA [32]. Therapeutically, DSA plays a vital role in catheterization procedures.

SPECT and PET image physiology and may serve as a surrogate in differential diagnosis and hemodynamic assessment. SPECT’s limitations are lack of anatomy, low resolution, and a relatively long acquisition time. PET is quantitative, and it has been established as the gold standard for the definition of the infarct and penumbra, but it is not employed in daily clinical practice because of its high cost, lack of anatomy, long acquisition time, and limited availability [26].

The stroke imaging modalities discussed above are summarized in Table 1.

## 3. Taxonomy of Stroke Image Processing and Analysis

Numerous image processing and analysis methods have been employed in stroke diagnosis and treatment, and their taxonomy is shown in Figure 2.

We divide the stroke image processing methods into two main groups, non-atlas/template-based methods and atlas/template-based methods. The non-atlas/template-based methods are further subdivided into five categories: intensity and contrast transformations, local segmentation-related, anatomy-guided, global density-guided, and artificial intelligence and deep learning AI/DL-based. The atlas/template-based methods, in turn, are subdivided into intensity templates and atlases with three atlas types: anatomy atlases, vascular atlases, and lesion-derived atlases.

### 3.1. Non-Atlas/Template-Based Methods

The non-atlas/template-based methods are based on various principles, and consequently are very heterogeneous. The simplest methods employ windowing for intensity and contrast transformations. The numerous lesion segmentation methods utilize a variety of techniques such as thresholding, region growing, edge detection, clustering, textures, watersheds, and wavelets, among others. The AI/DL-based techniques have been growing rapidly and include many approaches including linear regression, support vector machines, decision trees, random decision forests, k-nearest neighbors, k-means clustering, hidden Markov model, artificial neural networks, and convolutional neural networks with various architectures. Many of the non-atlas/template-based methods focus on local changes caused by a stroke lesion. A different approach is taken in the global density-guided methods exploiting density sampling over the entire image volume in numerous ranges. To increase the performance, many approaches are hybrid with a dominant technique. It should also be noted that the imaging properties of ischemic infarcts and hematomas are different, so the handing of these two types of stroke lesions requires different approaches.

The practical usefulness of these methods depends on their performance. Unfortunately, many of them, briefly outlined below, are not validated (especially on diverse and multicenter data) at all or the validation is on small samples. Even if validated, the validation is on various datasets and often with different performance measures. Consequently, the complete assessment and comparison of the methods presented below are difficult, if possible at all.

#### 3.1.1. Intensity and Contrast Transformations

The intensity and contrast transformation methods globally transform the image intensity and/or contrast. An example of a simple transformation of image brightness and contrast was presented by Mainali et al. [33], where the change in the window center and width level in standard windows and the introduction of improved stroke windows significantly improved the detection of early ischemic changes from 18% to 70%.

#### 3.1.2. Local Lesion Segmentation

Numerous methods have been proposed for infarct and hematoma segmentation from CT and MR images by employing a variety of techniques, including, among others, thresholding, region growing, edge detection, clustering, textures, watersheds, wavelets, rule-based expert systems, classification, and a combination of these [34,35,36,37,38,39,40,41,42,43,44,45,46,47,48,49,50,51,52,53,54,55,56,57,58,59,60,61,62].

##### Infarct Segmentation from CT Scans

A rule-based method to segment and label ischemic stroke lesions from NCCT was proposed by Matesin et al. [34]. The method determines a head symmetry axis based on moments, performs seeded region growing to identify multiple regions with uniform brightness, and executes rule-based region labeling using an expert system. The rules applied to identify the background, skull, brain, and cerebrospinal fluid are neighborhood- and intensity-based, and those for an ischemic lesion are symmetry-based.

A method based on the contouring of an ischemic stroke region in NCCT was proposed by Meilunas et al. [35]. The method performs slice filtering, smoothing, and extension of the stroke region boundary followed by infarct volume calculation.

A texture-based method with an unsupervised classifier was proposed by Usinskas et al. [36]. The method uses 18 unified textural features to segment an ischemic stroke region from NCCT images, such as joint features from the mean, standard deviation, histogram, and gray-level co-occurrence matrix.

A wavelet-based method was proposed by Przelaskowski et al. [37] based on the observation that data denoising and local contrast enhancement in a multiscale domain improve infarction perception. By applying wavelet-based image processing, enhancing the subtlest signs of hypodensity that otherwise were often invisible in a standard CT scan review, the sensitivity of ischemic lesion detection was increased from 12.5% for a standard CT scan preview to 56.3%.

A classification-based method in the intensity and wavelet domains was presented by Chawla et al. [38]. By comparing the cerebral hemispheres, the method detects and classifies a stroke-related abnormality into acute infarct, chronic infarct, and hemorrhage. The method involves three steps: image enhancement and denoising, detection of a brain symmetry line, and classification of abnormal slices. A two-level classification scheme applies an intensity histogram-based comparison for the identification of chronic and hemorrhagic cases as well as wavelet energy-based texture information for acute infarct detection.

A texture-based method using circular adaptive regions of interest was proposed by Tang et al. [39]. This method involves preprocessing (a threshold-based bone and artifact removal), creation of circular adaptive regions of interest, for each region locating a corresponding circular region on the other side of the brain image by reflection, and comparing each pair of the circular regions with several texture attributes. These attributes are calculated based on a gray-level co-occurrence matrix, and they include energy, entropy, inertia, inverse difference moment, correlation, prominence, shade, and variance. The method was tested on 10 acute and 10 chronic ischemic stroke cases and resulted in an accuracy of 86.96%.

A region growing method for infarct volume measurement in follow-up NCCT scans was presented by Boers et al. [40]. Following an interactive placement of the seed point in the infarcted area, the region growing was repeated for different thresholds in the range of 1.5–4.5 HU (Hounsfield units) with a step of 0.5 HU, resulting in seven segmentations. The region growing was limited by the midline to avoid leaking into the contralateral hemisphere.

A classification-based method to detect and segment ischemic lesions in NCCT was proposed by Vos et al. [41]. The method involves pixel classification and lesion candidate localization employing a Bayes classifier, segmentation of the candidate lesions and feature extraction via analyzing regional statistics, and combination of the extracted features into a likelihood of ischemia using a supervised classifier.

An unsupervised feature perception enhancement method for ischemic stroke detection on NCCT was presented by Tyan et al. [42]. The method performs preprocessing (by contrast enhancement), brain tissue extraction (by thresholding, blurring, and morphological operations), meaningful area extraction (via edge detection to identify the stroke area and unsupervised region growing followed by brain area partitioning into eight regions), and infarct regional location (by computing the brightness in these eight regions, determining areas with the smallest values compared to their counterparts, and analyzing their mutual relationships). The method produced an increased stroke diagnosis sensitivity of 83% in comparison to 31% for conventional diagnostic images.

A method based on textural analysis was proposed by Ray and Bandyopadhyay [43]. The method involves preprocessing (by noise and artifact removal), segmentation (by image gradient magnitude watershedding followed by thresholding), and feature extraction (by dividing an image into four quadrant regions, calculating for each region first-order texture measures (the mean, standard deviation, variation, kurtosis, skewness, and entropy), and selecting an abnormal region based on their values.

##### Infarct Segmentation from MR Scans

A semi-automated method of segmentation of DWI by employing adaptive thresholding and a Markov random field to model pixel adjacency and determination of the infarct volume was presented by Martel et al. [44].

A method combining nonlinear diffusion scale-space and geometric deformable models to segment ischemic stroke lesions from MR images was proposed by Weinman et al. [45].

A multimodal Markov random field model for multimodal MR segmentation of ischemic stroke lesions was proposed by Kabir et al. [46]. It simultaneously employed three MR sequences-T2W, FLAIR, and DWI.

A supervised approach applying semi-automatic segmentation of DWI and PWI scans and calculation of the perfusion–diffusion volume mismatch was presented by James et al. [47].

An unsupervised method for DWI infarct segmentation was proposed by Li et al. [48]. The method involves image preprocessing, calculation of tensor field and measurement of diffusion anisotropy, segmentation of infarction volume based on adaptive multiscale statistical classification, and partial volume voxel reclassification.

An automatic and rapid method to identify the infarcted slices and the hemisphere in DWI scans was proposed by Gupta et al. [49]. The method exploits the difference in percentile characteristics of intensity-normalized images. The method demonstrated average sensitivity and specificity for slice and hemisphere identification of 98.1% and 51.4%, and 91.7% and 91.7%, respectively.

A method for the detection of infarct lesions from anatomical MR scans was proposed by Shen et al. [50]. The method involves intensity-based segmentation by the fuzzy C-mean algorithm and the spatial location of tissues by prior tissue probability maps.

A method for the detection of ischemic stroke from MR images was presented by Karthik et al. [51]. It exploits discrete curvelet transformation to extract features on multiple scales and calculates statistical features.

A method for subacute ischemic stroke lesion segmentation in multimodal MR images was proposed by Maier et al. [52]. The method is based on extra tree forests and employs intensity-derived image features. The performance of the method measured by Dice’s similarity coefficient (DSC) [53] was 65%.

A method for automatic identification of stroke lesions in T1W images using naïve Bayes classification was proposed by Griffis et al. [54].

A method for the detection of ischemic stroke in MR images was proposed by Subudhi et al. [55]. The method exploits Delaunay triangulation and fractional order Darwinian particle swarm optimization and applies statistical and morphological features.

Bhanu Prakash et al. proposed a divergence measure-based method for infarct segmentation from DWI images by employing as divergence the ratio of the intensity probability density functions [56]. The method, evaluated on 57 datasets, yielded sensitivity, specificity, and DSC of 86.4%, 99.8%, and 72%, respectively.

##### Hematoma Segmentation

The imaging properties of ischemic infarcts are different from those of hematomas, and consequently, several dedicated methods for hematoma segmentation in hemorrhagic stroke have been proposed. Hematomas on NCCT vary in shape, size, location, density (intensity), contrast, and texture. Therefore, segmentation of hematomas depends on many factors, including partial volume effect (volume averaging), fuzzy and low contrast borders, noise, beam hardening, motion artifacts, hematoma closeness to bone, head tilt, and incomplete head coverage. The characterization of the distribution of intraventricular and intracerebral hematomas in NCCT scans was studied by Nowinski et al. [57]. This study used 289 serial retrospective scans and provided the quantitative relationship of hematomas with respect to gray and white matters.

Bhanu Prakash et al. proposed a modified distance regularized level set evolution method for hemorrhage segmentation from NCCT [58]. The method performs preprocessing, including filtering and skull removal, and segmentation employing a distance regularized level set evolution with shrinking and expansion. The method was validated on 200 NCCT scans collected at 10 different hospitals within the Clot Lysis Evaluating Accelerated Resolution of Intraventricular Hemorrhage (CLEAR III) multicenter, randomized phase III clinical trial. The scans were grouped as small, medium, and large based on the volume of blood, yielding a median DSC of 89.7% for the small group, 85.8% for the medium group, and 91.7% for the large group.

Patel et al. employed a 3D convolutional neural network for the segmentation and quantification of intracerebral hemorrhage in NCCT using a combination of contextual information on multiple scales [59]. The method was evaluated on two datasets of 25 and 50 patients, yielding a median DSC of 91% and 90%, respectively.

A texture-energy method for hemorrhage segmentation from NCCT scans was proposed by Bhanu Prakash et al. [60]. The method involves windowing, skull stripping, convolution with textural energy masks, and segmentation with combined thresholding and fuzzy C-means. The method, validated on 201 NCCT scans (from the CLEAR III phase III clinical trial), yielded median sensitivity, specificity, and DSC of 84.9%, 99.9%, and 87.1%, respectively.

An automatic segmentation approach based on machine learning employing a voxel-wise random forest classification was proposed by Scherer et al. [61]. Postprocessing of prediction maps involved the identification of the predominant clot, removal of isolated islands/voxels by morphological operations, and Gaussian smoothing of the probability map boundaries. The algorithm’s agreement with the manual reference tested on 30 NCCT scans was strong, with a concordance correlation coefficient of 0.95.

Bhanu Prakash et al. [62] compared three methods for intraventricular and intracerebral hemorrhage segmentation based on thresholding, clustering, and graph theory modified using textural energy-based normalization along with preprocessing (filtering and skull stripping) and postprocessing (artifact removal). The methods were tested on 201 NCCT scans (from CLEAR III). The median sensitivity, specificity, and DSC were 86.19%, 99.94%, and 86.55% for the modified thresholding; 83.23%, 99.93%, and 84.10% for the modified fuzzy C-means; and 87.28%, 99.81%, and 79.17% for the modified normalized cut method. The preprocessing and postprocessing enhanced the DSC by 10% and 3%, respectively. The use of textural energy along with the Hounsfield value in the modified methods increased the DSC by about 8–10%.

#### 3.1.3. Anatomy-Guided Methods

A few methods exploit neuroanatomy knowledge to assist in decision-making. An anatomy-enhanced method to identify potential areas of acute ischemia for middle cerebral artery (MCA) territory stroke was presented by Maldijan et al. [63]. The method first requires image preprocessing, including interpolation, scalp striping, and normalization, followed by the segmentation of two structures, one subcortical, the lentiform nucleus, and one cortical, the insula. Then, voxel densities in the segmented lentiform nucleus and insula of one hemisphere are compared with those in the contralateral hemisphere by applying the Wilcoxon two-sample rank-sum test.

The ASPECTS (Alberta Stroke Program Early CT Score) for stroke detection was proposed by Barber et al. [64]. It systemizes the detection and reporting of ischemic stroke by the visual identification of an ischemic hypodensity limited to the MCA territory subdivided into ten neuroanatomical regions, four subcortical and six cortical, which are located on two different axial CT slices. At present, this score is commonly accepted and employed in various methods.

#### 3.1.4. Global Density-Guided Methods

A standard way of stroke image processing is to detect and quantify local changes caused by the infarct or penumbra. In NCCT scans, the infarct occupies the density range between cerebrospinal fluid and white matter; however, it also causes the redistribution of density globally in the infarcted hemispheres. This global density redistribution, which might hardly be observable by the human eye, is encapsulated by the stroke imaging marker (SIM) [65]. The SIM provides the infarct spatial range in the axial, coronal, and sagittal orientations by statistically comparing numerous cumulative density distributions computed for the whole normal and infarcted cerebral hemispheres.

A SIM-based method was proposed and validated by Nowinski et al. [65] for fast and automatic detection, localization, and volume assessment of ischemic infarcts, including hyperacute, acute, lacunar, and chronic infarcts, as well as infarcts with hemorrhagic transformation and leukoaraiosis. The method identifies the patient-specific midsagittal plane [66], calculates density ranges of cerebrospinal fluid, white matter, and gray matter [67], computes the SIM for 54 density subranges, and determines two bounding boxes, the inner and the outer, localizing the infarct spatially. It also statistically calculates the volume of the infarct without explicitly segmenting it.

The method was quantitatively validated on 576 multicenter clinically confirmed strokes. The scans contained 322 “pure” acute ischemic infarcts (i.e., without any other noticeable pathology), 36 lacunar infarcts, 17 hemorrhagic transformations, 104 ischemic infarcts jointly with chronic infarcts, and 70 acute ischemic infarcts along with leukoaraiosis. The method matched 100% of experts’ infarct detection, yielding 99.8% inner localization specificity and 93.3% outer localization sensitivity when chronic infarcts, leukoaraiosis cases, and infarct volumes < 2 cm^3^ were excluded. For all the cases without infarct volumes < 2 cm^3^, the detection accuracy decreased to 95.7%. For all the cases, this detection accuracy was further reduced to 83.2%. Moreover, the method also detected 21 early ischemic infarcts (≤3 h) of which 15 were overlooked by stroke neuroradiologists.

This method was subsequently modified to cover a wider and finer spectrum of 168 density ranges and applied for hyperacute ischemia [68].

#### 3.1.5. AI/DL-Based Methods

Machine learning is a form of AI aiming to develop algorithms that allow computers to automatically learn from data without explicit programming. Machine learning methods are divided into supervised learning and unsupervised learning. In supervised learning, algorithms are first trained using some existing “ground truth” or “gold standard”. In ischemic stroke image processing, this is a collection of brain scans classified into infarct scans versus no-infarct scans. Unsupervised learning attempts to discover previously unknown classes, patterns, and/or structures in the data without known classification or previous training.

Supervised learning methods include, among others, linear regression, support vector machines, decision trees, random decision forests, and the k-nearest neighbors (k-NNs) algorithm, and the classifiers for ischemic stroke lesion segmentation were reviewed and compared by Maier et al. [69]. Unsupervised learning methods comprise, among others, k-means clustering, mixture models, and a hidden Markov model. Presently in radiology, the dominant type of machine learning algorithm is the artificial neural network (ANN) which is a cluster of interconnected nodes [70]. An ANN with multiple layers of interconnected nodes with representation learning is named deep learning. Deep learning has recently become the primary form of machine learning (ML) because of a convergence of theoretic advancements, openly available software, and hardware with sufficient computational power [71]. Deep learning via computationally efficient convolutional neural networks (CNNs) suits imaging [71]. CNNs require a significant amount of training data to avoid overfitting, and once the network parameters have converged, an additional training step is executed to fine-tune the network weights. In order to reduce the amount of training data and to generate a more accurate segmentation of biomedical images, U-Net architecture was proposed by Ronneberger et al. based on CNNs [72].

A stroke detection method using texture features combined with various ML algorithms was proposed by Rajini and Bhavani [73]. The method performs preprocessing, segmentation, brain midline tracing, extraction of texture features (employing a gray-level co-occurrence matrix between the left and right hemispheres), and classification by a binary classifier. By applying a support vector machine, k-nearest neighbors, ANN, and decision tree classifiers, the method yielded an accuracy of 98%, 97%, 96%, and 92%, respectively.

An infarct segmentation method employing CNN DL was proposed by Sales Barros et al. [74]. The method employs preprocessing to segment the intracranial region (via region growing and morphological operations) and CNN-based infarct segmentation (by means of the CNN architecture having two convolutional layers followed by two fully connected dense layers, each dense layer with 256 nodes). For validation, 570 NCCT stroke scans were used for training, 60 for parameter fine-tuning, and 396 to test segmentation, achieving a DSC of 18%. To improve segmentation, the scans were additionally divided into three infarction classes with fixed thresholds: severe {14, 22} HU, intermediate {22, 32} HU, and subtle {32, 44} HU. Subsequently, using three CNNs, the obtained values of the DSC were 78% for the severe class, 61% for the intermediate class, and 37% for the subtle class.

A DL method aiming to automate the ASPECTS score based on texture features extracted from each ASPECTS region to train a random forest (RF) classifier was proposed by Kuang et al. [75]. The method initially achieved a sensitivity of 66.2% and specificity of 91.8%, which were further improved by ASPECTS dichotomization (>4 and ≤4), giving a sensitivity of 97.8% and specificity of 80%.

An infarct segmentation method combining ML exploiting a cascaded RF and interactive segmentation was presented by Kuang et al. [76]. The method executes expert initialization, RF learning and classification with a two-stage training and testing classifier, and optimization-based segmentation. The cascaded RF learning is applied to classify each voxel into normal or ischemic and to compute an infarct probability map. Four features are extracted: intensity, statistical information in a local region, the symmetric difference compared to the contralateral side, and the spatial probability of infarct occurrence. These features train the first-stage RF classifier whose coarse segmentation results are used to train the second-stage fine classifier, providing an estimation of the infarct probability map. After optimization, this map gives the final segmentation. The method achieved a DSC of 79% and considerably outperformed some other AI methods, including the RF-based methods and CNN-based U-Net.

A method based on a dense multi-path contextual generative adversarial network (MPC-GAN) was proposed by Kuang et al. [77]. The method applies a dense multi-path U-Net as a generator regularized by a discriminator network. The generator and discriminator input contextual information including bilateral intensity difference, infarct location probability, and distance to cerebrospinal fluid. The MPC-GAN yielded a DSC of 72.6% and outperformed some state-of-the-art segmentation methods, such as U-Net, U-Net-based GAN, and RF-based segmentation.

A DL-based semi-D-Net method for the simultaneous segmentation of infarcts and hemorrhages was presented by Kuang et al. [78]. The method combines network-learned semantic information, local image context, and a user-initialized prior to a multi-region contour evolution scheme globally optimized by a convex relaxation technique. The method yielded mean DSCs of 67.4% for ischemic infarct, 65.3% for hemorrhage, and 72.5% for both and outperformed other DL methods, such as U-Net, D-Net, demi-RF, and semi-U-Net.

A probabilistic neural network and adaptive Gaussian mixture model employed to segment DWI infarcts was proposed by Prakash et al. [79]. The method yielded an average DSC of about 60%, and an average sensitivity and specificity of 81% and 99%, respectively.

A method for automatic acute ischemic lesion segmentation from DWI using CNNs was proposed by Chen et al. [80]. This approach involves two CNNs, an ensemble of two DeconvNet and multiscale convolutional label evaluation net, and yielded a mean DSC of 67%.

A DL method for acute and subacute stroke lesion segmentation employing multimodal MR anatomic, DWI, and PWI imaging was proposed by Clèrigues et al. [81]. The method involves preprocessing based on hemispheric symmetry and uses a U-Net-based CNN architecture.

A DL segmentation with a U-Net model combined with variational mode decomposition as a preprocessor was employed to segment infarct lesions from T1W scans by Paing et al. [82]. The method achieved an average DSC of 66.8%.

A method for ischemic lesion segmentation by employing an ensemble of multiscale region-aligned CNNs was proposed by Karthik et al. [83]. It yielded a mean DSC of 77.5%.

DL was employed to predict final ischemic stroke lesions from initial MR imaging by Yu et al. [84].

DL-based detection and segmentation of diffusion abnormalities in acute ischemic stroke along with classification of the DL models were presented by Liu et al. [85]. The developed public tool with deep learning networks was tested on over 2300 DWI scans.

A supervised learning method for recognizing stroke and transient ischemic attack and differentiating them from stroke mimics in an emergency setting was proposed by Albedi et al. [86]. The method employs an ANN model utilizing the Neuralnet software package. The method, tested on 260 patients, yielded 80% sensitivity and 86% specificity.

To predict the ischemic core and determine the optimal threshold using CTP, an ANN model was designed by Kasasbeh et al. [87] and clinical parameters integrated into it, including NIH (National Institutes of Health) Stroke Scale (NIHSS), sex, age, and time from symptom onset to CTP. For the optimal threshold, the sensitivity for predicting the ischemic core was 90% and the specificity 62%.

An ML approach to classify time since stroke onset from MR images and perfusion maps was proposed by Ho et al. [88]. The approach employs five ML methods: logistic regression, random forest, gradient boosted regression tree, support vector machine, and stepwise multilinear regression. The results demonstrate that the best machine learning model can outperform the current state-of-the-art DWI–FLAIR mismatch method (see Section 4) for determining eligibility for thrombolytic therapy in cases of unknown time since stroke onset.

A multiparametric diffusion MR imaging method to train ensembles of CNNs was presented by Winzeck et al. [89]. By employing a combination of three diffusion maps, DWI, ADC, and low b-value, a median DSC of 80.2% was achieved, yielding results comparable with manual lesions delineated by experts.

A multi-path 2.5-dimensional CNN system for segmenting stroke lesions in MR scans was proposed by Xue et al. [90]. It employs nine U-Nets and yielded a mean DSC of 54%.

A DL model to identify acute infarct on MR scans was proposed by Bridges et al. [91]. The neural network architecture employed was based on the 3D U-Net segmentation model. The model trained on over 6600 MR studies yielded a median DSC of 77.6%.

### 3.2. Template- and Atlas-Based Methods

The commonality of the template-based and atlas-based methods is that each group requires a mapping of a template or an atlas to a patient’s specific scan or map. The difference between them is that the intensity-based template exploits image intensity for the whole brain scan, whereas the brain atlases utilize domain knowledge, primarily neuroanatomy, or distribution of stroke lesions and the processing is region of interest (ROI)-based.

#### 3.2.1. Template-Based Methods

An intensity template-based method aiming to delineate infarcts and hemorrhages was proposed by Gillebert et al. [92]. It employs a normal CT scan template created from 72 non-stroke subjects for defining areas with hypo- and hyperintense signals. The method involves patient scan preprocessing (via threshold-based clustering, intensity transformation, space normalization, isotropic reslicing, and smoothing) followed by statistical analysis for lesion detection by a voxel-by-voxel comparison of the preprocessed scan with the intensity template. The performance of the method measured by the DSC was between 52% and 89% depending on the level of thresholding and the degree of smoothing.

#### 3.2.2. Atlas-Based Methods

##### Methodology

The atlas-based methods can be considered an extension of the anatomy-guided methods for a higher number of structures delineated automatically. Generally, these methods have two major steps: (1) mapping of the atlas to a patient-specific scan and (2) atlas-assisted analysis of a scan or a perfusion map. Atlas-to-scan mapping is achieved by applying spatial registration methods that produce an individualized atlas, and some of these methods have been reviewed in [93,94]. Analysis of the scan or map with the individualized (superimposed) atlas is application-dependent. Atlas-assisted stroke image analysis is ROI-related, and three types of ROIs have been distinguished in stroke image management: atlas-defined ROIs, atlas-quantified ROIs, and ROIs creating an atlas [95]. An ROI can be defined either by atlas-guided anatomy and/or blood supply territories or by scan-derived pathology. The brain atlas can be exploited either to define an ROI or to quantify it.

The atlas-defined ROI is a region in the scan delineated by the superimposed individualized atlas on it, such as the thalamus, postcentral gyrus, or Brodmann’s area 17. The atlas-quantified ROI involves a specific region (typically a lesion) or regions outlined in the scan that is/are subsequently quantified by the individualized atlas or atlases. This quantification provides a list of all atlas structures overlapping with the considered ROI, and for each structure, its quantitative contribution to this ROI is determined in terms of volume and percentage of occupancy. The ROIs creating an atlas aggregate the content of all the considered ROIs across the patient population to form a probabilistic atlas.

To manage acute stroke, a rapid and automated atlas-to-scan mapping is required. To automatically process MR scans, a dedicated method has been developed based on the fast Talairach transformation proposed by Nowinski et al. [96] by employing the modified Talairach landmarks [97]. To handle CT scans, another fast and automated method has been developed based on ellipse fitting [98].

##### Atlases

The brain atlases applied in stroke management belong to three categories, namely, anatomy, vasculature, and lesion-based.

An example of an atlas of anatomy is the *Cerefy* brain atlas [99]. It is derived from the classic Talairach–Tournoux brain atlas [100], which contains gross anatomy in axial, coronal, and sagittal orientations. The *Cerefy* brain atlas is electronic, deformable, fully parcellated, labeled, and placed in a stereotactic coordinate system. The construction of this atlas and its continuous enhancements were addressed in [101,102].

Vascular atlases can contain blood supply territories and/or cerebral vessels. An example of an atlas of blood supply territories is presented in [103], created by the parcellation of the atlas of anatomy into the blood supplying territories. In this way, both atlases are by definition in spatial correspondence, and the individualization of the atlas of anatomy results in the individualization of the atlas of blood supply territories. The atlas of blood supply territories has at least three applications, as it assists in the automated calculation of thrombolysis conditions (discussed in Section 4 and Section 5.2), provides a stroke subtype, and increases the number of ROIs in atlas-assisted analysis.

A 3D stroke atlas [104] contains the 3D vasculature with simulated vascular lesions and correlates disorders with the underlying vascular neuroanatomy by linking a cerebrovascular lesion location with the resulting disorder along with the corresponding signs, symptoms, and/or syndromes.

The Population Stroke Atlas (PSA) belongs to the lesion-derived atlas category [105]. It aggregates data from previously managed patients and applies them to the currently treated patients. The PSA links neurological examination parameters with pathology localized on neuroimages for a population of stroke patients. These neurological parameters include history, demographics, hospitalization, laboratory parameters, clinical measures, and outcomes, among others. In particular, in [105], the PSA was calculated for sex, survival, NIHSS, Barthel Index at 30, 90, 180, 360 days, modified Rankin Scale at 7, 30, 90, 180, and 360 days, white blood cell count, C-creative protein, glucose at the emergency department, history of hypertension, and history of diabetes, among many others. The PSA aggregates a multiplicity of parameters and presents the distribution of each parameter in the infarct region as a 3D image volume. The simplest means of aggregation is to assign the parameter value to all the voxels within the delineated infarct and accumulate them across all cases, which results in a spatial distribution of the average value of a considered parameter. Generally, the process of parameter value aggregation is more sophisticated and takes into account the size, overlap, and distance of the infarcts forming the PSA and that (or those) of the treated patient by weighting their contributions and selecting for the PSA calculation’s most relevant cases. The calculated PSA 3D volumes can be further processed, analyzed, and visualized, as well as knowledge, trends, and predictions extracted from them.

Another stroke atlas in this category is a probabilistic atlas [106] created from 22 cases providing a spatial distribution of acute infarcts. This atlas may be considered a special case of the PSA for image intensity only and without weighting and case selection.

To quantify the impact of infarct location on stroke severity, Menzes et al. [107] created a brain atlas containing the location-weighted values from 80 ischemic stroke patients. The predefined anatomical regions, though without infarcts included, were weighted depending on their size and the NIHSS value.

A deformable raw atlas is applicable in stoke image processing; however, it is more useful and efficient to embed such an atlas into a CAD (computer-aided detection/diagnosis/decision-making) system dedicated to stroke management and to provide suitable support for the performed procedure [108]. Below in Section 5, we provide some illustrative examples of stroke CAD systems.

## 4. Taxonomy of Stroke Treatment

The key therapeutic goals in acute stroke therapy are to restore blood flow to the affected area via recanalization and to diminish the influence of ischemia on tissue. Most of the decision criteria are derived from neuroimaging. A diagram illustrating the taxonomy of stroke treatment along with imaging modalities employed and decision criteria applied is presented in Figure 3.

The stroke treatment procedures are first divided with respect to stroke location, namely, arterial stroke and venous stroke. Most strokes occur when a thrombus (clot) occludes an artery supplying the brain. Then, treatment is by means of thrombolysis or fibrinolysis, which consist in delivering a thrombolytic agent to dissolve a thrombus. The main thrombolytic agents include streptokinase, urokinase, and tissue plasminogen activator (t-PA), the last having the best safety profile [10]. However, the risk of thrombolysis is that it may lead to hemorrhage, and certain treatment guidelines are recommended [109,110].

Thrombolysis can be performed intravenously (IVT) or intraarterially (IAT), or by combining both procedures. IVT is the standard treatment of acute ischemic stroke for vessel recanalization approved in patients up to 4.5 h after stroke symptom onset. IVT advantages are technical simplicity, wide availability, and capability for rapid initiation.

IAT delivers the thrombolytic agent directly to the clot via a microcatheter. This therapy permits the delivery of a smaller dose at a higher local concentration, thereby reducing the risk of potential complications such as intracranial hemorrhage and improving the rates of recanalization. It also extends the treatment window up to 6 h [111]. The combined IVT–IAT procedure was demonstrated to have better recanalization rates than IVT alone [112].

Thrombolysis requires checking of four main conditions through imaging, namely, the absence of hemorrhage; the presence and site of the vessel occlusion; the presence, location, and size of the infarct; and the presence and size of the penumbra. The main contraindication of thrombolysis is hemorrhage, and its absence can be effectively confirmed by NCCT or MR T2* imaging. After ruling out hemorrhage, it is crucial to determine whether confirmed ischemia is present and to assess its extent. The volume of the infarct is of critical prognostic importance, as it is one of the highest-ranking independent predictive signs [113]. An infarct volume of 70–100 mL in the MCA territory is highly specific for poor outcomes, independently of the existence of penumbra or a successful vascular recanalization [114]. Note that the risk of significant hemorrhage during treatment increases in proportion to the size of the infarcted tissue, especially if the volume is greater than 100 mL [115]. In addition, the ratio of the infarct volume to that of the MCA territory predicts hemorrhagic transformation when higher than, depending on different authors, one-half [116] or one-third [117]. The size of the MCA territory is typically estimated visually or it can be calculated automatically in 3D by employing a brain atlas [103] (see also Section 5.2).

The detection and site of occlusion, if any is present, is also important using CTA and MRA.

The detection of the penumbra and the evaluation of its size is vital, as it is the main target for reperfusion therapies. Having evaluated the size of the infarct and the penumbra, another critical factor is the presence of a mismatch. From a stroke onset standpoint, there are two situations, the time from stroke onset is known or this time is not clear, for instance, in acute stroke on awakening.

For the known time from stroke onset, a PWI–DWI mismatch is employed that is calculated as the penumbra-to-infarct volume ratio. The penumbra is determined from the perfusion maps. Within the infarct, the loss of blood flow is total and both CBF and CBV are decreased. In the penumbra, only CBF is decreased, whereas self-regulation keeps CBV within normal limits. In both infarct and penumbra, MTT is increased because the contrast agent circulates with difficulty via the capillary bed of the compromised tissue. The PWI–DWI mismatch helps identify acute stroke patients who can benefit from reperfusion therapy. In several stroke trials, a PWI–DWI mismatch ratio of >1.2 has been widely used [118,119,120]. The optimal PWI–DWI mismatch has been studied by Kakuda et al. [121] and the highest sensitivity and specificity were yielded by a mismatch ratio of 2.6.

When the time from stroke onset is unknown, the DWI–FLAIR mismatch is used, which is defined as the presence of visible acute ischemic lesions on DWI with no noticeable hyperintensity in the corresponding area on FLAIR images. This mismatch is caused by the fact that the ischemic tissue is visible on DWI as early as 35 min after the onset of stroke symptoms, whereas it takes 3–4 h for the ischemic tissue to appear in FLAIR. The DWI–FLAIR mismatch is the current state-of-the-art approach to provide clinicians with insight into stroke onset time. It also facilitates the differentiation of stroke from non-strokes and stroke-mimicking conditions.

IVT and IAT are pharmacologic procedures where the lysis of a thrombus takes a certain time; besides, some clots may be resistant to the thrombolytic agent, e.g., thrombi longer than 6 mm [10]. Consequently, thrombolysis yields lower recanalization rates than mechanical thrombectomy [10]. Mechanical thrombectomy consists in a mechanical removal of the clot by entering the vessel via a microcatheter directly to the position of the clot and then removing it by either retrieving it distally or by proximal aspiration. This therapy expedites recanalization and can be used in cases when pharmacologic IV and IA thrombolysis were unsuccessful. A wide range of mechanical thrombectomy devices has been constructed, including distal clot removal devices (MERCI and Catch), stent retrievers (Solitaire, Trevo, and Revive), and an aspiration device (Penumbra). Mechanical thrombectomy has demonstrated efficacy and safety, even beyond a 6–8 h window out to 24 h [26].

Cerebral venous and sinus thrombosis (CVST) is a relatively rare cause of stroke. Depending on the site of thrombosis, three types of CVST are distinguished: sinus thrombosis, deep cerebral venous thrombosis, and cortical vein thrombosis. MR, in particular MRA and T2*, is considered the gold standard in CVST diagnosis, while treatment includes conservative therapy, pharmacologic thrombectomy, and mechanical thrombectomy [10].

The above stroke therapies serve for the treatment of ischemic stroke, which accounts for approximately 80% of cerebrovascular events [4]. There are also several therapies to treat hemorrhagic stroke, especially since the mortality rate of acute intracerebral hemorrhage can reach 40% [122]. One of them is to evacuate the hemorrhage surgically by performing craniotomy. This procedure additionally facilitates decompression. A less invasive approach is to insert a catheter into the ventricular system and lyse the blood clot by administering t-PA [123] (see also Section 5.3).

## 5. Stroke CAD Systems

There are a few CAD systems assisting stroke to a certain extent. A CAD system [39] aids in the early detection of ischemic stroke for small lesions. WebParc measures stroke lesion volume and provides lesion location with respect to a registered brain template [124]. A CAD system [125] enhances the accuracy of radiologists’ performance in the detection of lacunar infarcts on T1W and T2W images. A CAD system [126] characterizes atherosclerotic plaques by incorporating scores to determine the risk of plaque rupture and thus that of stroke. A CAD system [127] aids in the detection of small acute intracranial hemorrhages.

The brain atlases are useful as a core technology in CAD systems for stroke screening and stroke occurrence assessment [95] by quantifying white matter lesions that are associated with an increased stroke risk [128] and its recurrence [129], as well as in outcome prediction by employing, for instance, the PSA [105].

Below we feature in more detail three CAD stroke supporting systems that have been used in several hospitals on a trial basis.

### 5.1. NCCT Stroke CAD System in Emergency Room

As already discussed, the first-line diagnosis for evaluation of acute stroke in the emergency room (ER) is NCCT; however, its sensitivity to ischemia is very low, 25%, and within the first 3 h, it is lowered to 7%. Practically, these values can be even lower at the early assessment, as the scans are often viewed initially by non-stroke clinicians, such as emergency physicians, non-neuroradiologists, neurology or radiology residents, or even junior staff on duty, before being interpreted by stroke neuroradiologists [130]. Therefore, automated and fast detection, localization, and quantification of acute stroke infarcts could assist in enhancing and expediting its diagnosis in the ER.

An atlas-assisted stroke CAD system in the ER that exploits deformable atlases of anatomy and blood supply territories has been described by Nowinski [95]. It rapidly and automatically detects stroke, distinguishes between ischemic and hemorrhagic strokes, and localizes an infarct or hemorrhage. This CAD system performs a fast atlas-to-NCCT scan mapping by employing the ellipse fitting method [98], removes the skull and extracts the brain, segments the ventricular system to determine cerebrospinal fluid regions [131], and subsequently removes the cerebrospinal fluid from the extracted brain because its density range overlaps with that of infarcts. Then, the CAD system statistically analyzes the image density differences between the left and right hemispheres in numerous ROIs defined by the atlases. The simultaneous use of two brain atlases increases the number of ROIs. The atlas-assisted analysis exploits the atlas-defined ROIs by comparing all the corresponding left and right ROIs in both atlases. A diagnosis with lesion detection and localization is made based on statistical testing of multiple conditions, including mismatch between the density distribution curves calculated in the corresponding left and right ROIs.

### 5.2. MR Stroke CAD System for Thrombolysis

The standard treatment procedure for acute ischemic stroke is as discussed above thrombolysis. Intravenous thrombolysis is the preferred treatment for most ischemic strokes; however, uncontrolled, it may result in hemorrhagic transformation. Therefore, after excluding hemorrhage, the procedure is performed provided that certain thrombolysis conditions are satisfied, including (1) the size of the infarct relative to that of the MCA territory, (2) the presence of a diffusion–perfusion mismatch, calculated as the penumbra-to-infarct volume ratio, and (3) the location of vessel occlusion, if any. The ratio of the infarct volume to that of the MCA territory predicts hemorrhagic transformation when greater than one-half [116] or one-third [117].

The stroke CAD system developed by Nowinski at al. [103] supports thrombolysis by automating the calculation of the thrombolytic conditions (the presence of vessel occlusion is checked interactively in 3D) and extending the standard procedure by taking into account infarct and penumbra locations. For extracted infarcts from DWI scans and penumbrae from perfusion maps, this stroke CAD employs the atlases of anatomy and blood supply territories. The infarct and penumbra are considered ROIs that are quantitatively assessed by means of both atlases. This CAD system automatically analyzes the entire regions occupied by the infarct and the penumbra and calculates for them: (i) names of all anatomical structures and blood supply territories within each region; (ii) volume of occupancy for each structure and blood supply territory; and (iii) percentage of occupancy for each structure and territory. The system calculates the infarct-MCA territory ratio for DWI images and the penumbra-MCA territory ratio for perfusion maps in 3D, in contrast to the standard procedure of assessing the first ratio just on a single slice (or a few slices).

### 5.3. NCCT CAD System for Hemorrhagic Stroke

The CAD system for hemorrhagic stroke aims at the progression and quantification of blood clot removal over time. It segments, quantifies, and displays hematomas in 2D and 3D, and supports evacuation of hemorrhage via thrombolytic treatment by monitoring progression and quantifying clot removal [132]. The procedure requires a catheter to be stereotactically inserted into the ventricular system, t-PA administered through it, and a series of NCCT scans acquired to monitor the outcome of clot lysis [123]. This CAD system supports a seven-step workflow, including selection of patient, adding a new study, processing the patient’s scans, showing segmentation results, plotting hematoma volumes, showing 3D synchronized time-series hematomas along with their intraventricular and intracranial compartments, and generating a report. The key components of the CAD system architecture are the main application with the user interface, tools, and hematoma automated segmentation algorithms [58,60,62]. The tools include a contour editor to edit the segmented results, 3D surface modeler, 3D volume measure, histogramming, hematoma volume plot, and 3D synchronized time-series hematoma display and manipulation. The CAD system was employed in the CLEAR III and MISTIE III (Minimally Invasive Surgery plus rt-PA for Intracerebral Hemorrhage Evacuation) phase III, multicenter clinical trials [123].

## 6. Discussion

Acute ischemic stroke management is a very broad and dynamic field. Here we consider it from three critical perspectives: diagnostic and therapeutic imaging, image processing and analysis, and treatment. Each of these perspectives is categorized, and all three corresponding taxonomies are presented in the form of comprehensive and simple diagrams demonstrating relationships among their components.

Neuroimaging plays a crucial role, and the imaging methods in stroke management are well established. The most frequently employed acute stroke imaging modalities are CT and MR. MR is the most complete method to examine stroke patients, and it has higher sensitivity and specificity for diagnosis than CT. NCCT remains the first-line diagnosis for emergency evaluation of acute stroke because of its widespread availability, speed, and low cost. Advancements in stroke imaging are predominantly in higher spatial and temporal resolutions as well as speed of image acquisition. Moreover, the use of AI in image generation additionally reduces cost without compromising quality.

Two main procedures for acute stroke treatment are thrombolysis (IVT and IAT) and mechanical thrombectomy. IVT is the standard treatment of acute ischemic stroke for vessel recanalization because of its technical simplicity, wide availability, and capability for rapid initiation. It is approved in patients up to 4.5 h after stroke symptom onset, and this time window is extended in IAT to 6 h and mechanical thrombectomy to 24 h. Endovascular thrombectomy has transformed acute stroke management via improvements in devices and stroke workflows [133]. The most promising approach to the recanalization of intracranial vascular occlusions is bridging therapy through the initial IVT followed by intraarterial treatment.

Stroke image processing and analysis methods are divided into two main groups: non-atlas or template-based and atlas/template-based. The non-atlas/template-based methods are further subdivided into five categories: intensity and contrast transformations, local segmentation-related, anatomy-guided, global density-guided, and AI/DL-based. Out of all three taxonomies, the image processing and analysis taxonomy seems the least established, as new methods keep rapidly evolving and the clear boundaries between the groups of methods are sometimes indistinctive. For instance, many image processing and analysis methods employ supervised or unsupervised classification. Similarly, AI/DL methods usually increase performance with prior image processing. Therefore, many methods can be rather considered hybrid with some dominant techniques.

There are numerous methods developed for local infarct segmentation from CT and MR images and hematoma segmentation from CT images based on various techniques, such as thresholding, region growing, edge detection, clustering, textures, watersheds, wavelets, rule-based expert systems, classification, and a combination of these. Many of them are poorly validated on a few cases only or not validated at all, which hampers their evaluation and comparison.

Global density-guided methods exploiting density sampling over the entire volumetric image in various ranges were tested on numerous variable scans from four centers, yielding promising initial results, especially for early stroke detection in NCCT [65,68]. The CAD stroke system supporting these methods was trial-licensed to a few hospitals, but funding for clinical trials was not secured to advance this work.

The development of brain atlases is a dynamically expanding field [134] propelled by brain big projects [135], and the use of brain atlases in stroke management has several advantages in diagnosis, treatment, and prediction. Atlas-related operations are automated, including atlas-to-scan mapping and atlas-assisted analysis. These operations are also rapid, taking a few seconds, making them suitable to handle time-sensitive situations in stroke. The atlases provide a quantitative assessment of the infarct and penumbra in terms of the underling anatomy and blood supplying territories, estimate the size of the MCA and other territories in 3D, and enable an automated generation of ROIs. Note that the current practice is stroke lesion size-centric, while the brain atlas analysis also enables lesion location evaluation, which can aid in determining the functions at risk of being lost and contributing to a better decision-making process.

The use of AI and DL in acute stroke management has been growing rapidly, facilitated by a convergence of theoretic advancements, openly available software, and hardware with sufficient computing power. For instance, the number of AI acute stroke publications on PubMed grew from 54 in 1990 to 932 in 2023 (1990/54, 2000/210, 2005/313, 2010/439, 2015/675, 2020/958, and 2023/932). The number of AI acute stroke publications on Google Scholar is 1,830,000. Moreover, the world’s first, most advanced, and fully automated AI-based commercial solution for stroke evaluation, the Brainomix 360 stroke platform, has been developed [136].

The proposed classification for stroke image processing and analysis is not exhaustive and the categories are not inseparable. It can further be subdivided, for instance, based on the modality, as was illustrated for CT and MR in Section 3.1.2. Another criterion may be the type of vascular lesion, such as hyperacute ischemic infarct, acute ischemic infarct, subacute ischemic infarct, chronic ischemic infarct, transient ischemic attract, hemorrhage, and stroke mimics lesion. AI/DL methods can be further subdivided, e.g., based on machine learning method, kind of classifier, type of architecture applied, quality of data handled (complete data versus fragmented, noisy, small, and missing data), and dimensionality of architecture (note that most DL methods were developed in 2D architectures, ignoring the 3D contextual information, and only recently have 2.5D and 3D architectures been applied, e.g., [137]).

The proposed image processing and analysis taxonomy is neither unique nor final. Several authors proposed some other taxonomies for stroke image processing based on various criteria. Rekik et al. [16] divided methods for ischemic stroke image management into four groups: pixel- and voxel-based classification, image-based segmentation, atlas-based segmentation, and deformable model-based segmentation. Nowinski et al. [13] proposed a 2 × 2 matrix-based classification with local versus global image processing and analysis, and density versus spatial sampling. Inamdar et al. [17] divided methods in acute stroke diagnosis into non-ML/DL-based, ML-based, and DL-based techniques. The non-ML/DL-based methods include region growth, texture extraction in spatial and frequency domains, and contour-based. The ML-based methods contain three types: classification based on discriminants such as texture, density, and contour-based analysis; purely probabilistic; and hybrid. The DL-based techniques extensively explore the CNN architecture with 3D kernels with many modifications, including novel architectures, such as U-Net and ResNets.

The diagnosis and treatment of acute stroke have changed dramatically during the last few years because of the recent advances in endovascular treatment (as reviewed by Widimsky et al. [138] in terms of key stroke trials, devices, and techniques) and portable stroke imaging, which is crucial in time-critical situations. In particular, this includes the Mobile Interventional Stroke Team (MIST) model of stroke treatment that consists in traveling to perform endovascular thrombectomy in the hospital where the patient is first diagnosed. The MIST trial demonstrated that the use of a MIST was faster and led to improved discharge outcomes when compared with the drip-and-ship model, i.e., transporting the patient to the nearest primary stroke center [139]. Another novel concept is a portable ultrasound tomography device for vascular brain lesions proposed by Kazmierski [140]. This device contains multiple transcranial ultrasound probes placed on the surface of the head allowing for a significant reduction in time from stroke onset to thrombolytic therapy.

This review, particularly concerning the stroke image processing and analysis methods, is rather illustrative for the proposed taxonomies and far from being a complete review of this extensive field. The advances in acute stroke are tremendous and the number of papers published in this field is enormous. For instance, PubMed, checked on 24 February 2024, gives 446,026 results for the term “acute stroke”, 106,102 results for “acute stroke imaging”, 239,206 results for “acute stroke therapy”, and 10,189 results for “acute stroke image processing”. Google Scholar, checked on 24 February 2024, gives about 3,520,000 results for the term “acute stroke”, about 3,000,000 results for “acute stroke imaging”, about 3,030,000 results for “acute stroke therapy”, and about 1,140,000 results for “acute stroke image processing”.

## 7. Summary and Conclusions

Stroke is the most common life-threatening neurological disorder, a leading cause of death, and a main cause of permanent disability. It has a deep impact on public health and generates vast costs for primary treatment, hospitalization, rehabilitation, and chronic care. Stroke management is demanding because of the variety of diagnostic imaging modalities, image processing and analysis methods, and treatment procedures, protocols, and criteria. This work categorizes methods for stroke imaging, image processing and analysis, and treatment, and provides their taxonomies in the form of simple and comprehensive diagrams, as well as presents a state-of-the-art review illustrating the introduced taxonomies.

Imaging plays a vital role in stroke management, and the most frequently employed imaging modalities are CT and MR imaging. CT includes unenhanced non-contrast CT, the first-line diagnosis for acute stroke emergency evaluation, CT angiography, and CT perfusion. MR is the most complete method to examine stroke patients, with sensitivity and specificity for the diagnosis of ischemia in the early hours after stroke onset demonstrating normal and pathological anatomy, angiography, diffusion, perfusion, and spectroscopy. MR angiography is useful to evaluate the severity of artery stenosis, vascular occlusion, and collateral flow. Diffusion-weighted imaging is the gold standard for evaluating ischemia. MR perfusion-weighted imaging examines hemodynamic conditions at the microvascular level, enabling the evaluation of the penumbra. Other stroke imaging modalities, such as ultrasonography, digital subtraction angiography, positron emission tomography, and single-photon emission computed tomography, provide a surrogate role.

The stroke image processing and analysis methods are here divided into non-atlas/template-based and atlas/template-based. The non-atlas/template-based methods are subdivided into intensity and contrast transformations, local segmentation-related, anatomy-guided, global density-guided, and artificial intelligence/deep learning-based. The atlas/template-based methods are subdivided into intensity templates and atlases with three atlas types: anatomy atlases, vascular atlases, and lesion-derived atlases.

The stroke treatment procedures are divided with respect to stroke location into arterial stroke and venous stroke. Arterial stroke is treated by thrombolysis and mechanical thrombectomy. Thrombolysis can be performed intravenously (IVT) or intraarterially, or by combining both procedures. IVT is the standard treatment of acute ischemic stroke. Venous stroke is treated by conventional therapy, pharmacologic thrombectomy, and mechanical thrombectomy. The treatment taxonomy has additionally been extended with imaging modalities and decision-making criteria.

All three taxonomies have been widely illustrated, compared against other state-of-the-art classifications, and the role of the presented methods and decision-making criteria discussed within the processes of acute stroke management.

## Figures and Tables

**Figure 1 diagnostics-14-01057-f001:**
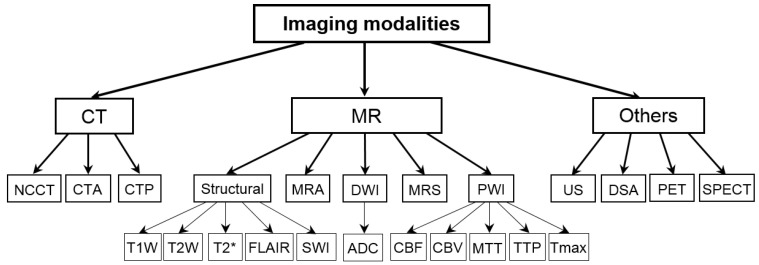
Diagram illustrating the taxonomy of stroke imaging modalities.

**Figure 2 diagnostics-14-01057-f002:**
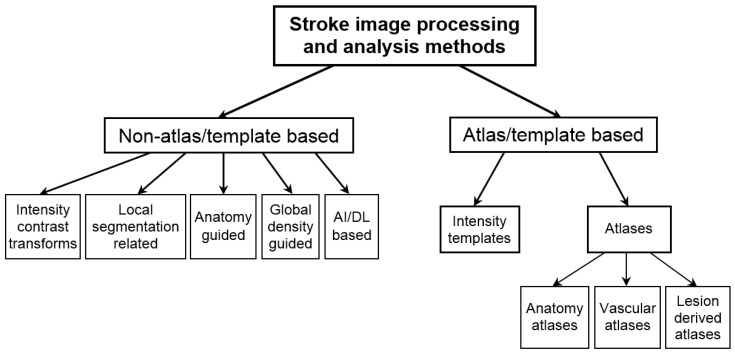
Diagram illustrating the taxonomy of stroke image processing and analysis methods.

**Figure 3 diagnostics-14-01057-f003:**
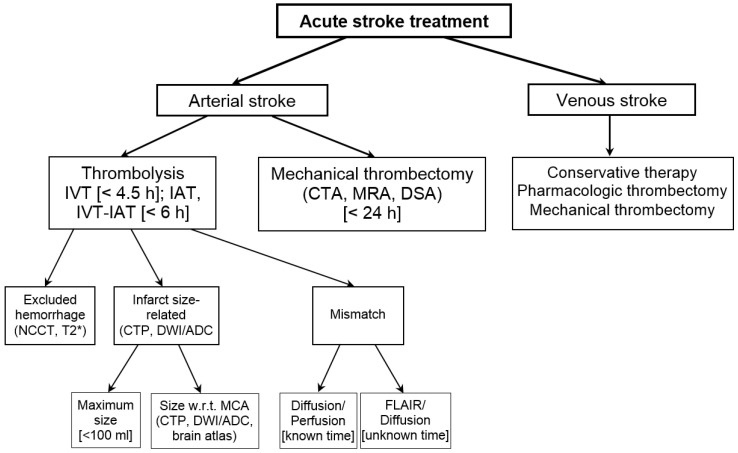
Diagram illustrating the taxonomy of stroke treatment along with imaging modalities (indicated in parentheses) and decision-making criteria (in brackets).

**Table 1 diagnostics-14-01057-t001:** Stroke imaging modalities.

CT	MR	Others
NCCT is the first-line diagnosis for emergency evaluation. CTA images cervical and cerebral arteries and identifies or excludes large-vessel occlusion, detects stenosis, and evaluates the collateral vascular network. CTP allows for the calculation of perfusion maps useful for the evaluation of infarct and penumbra.	T1W, T2W provide anatomy. T2* enables exclusion of hemorrhage. SWI reveals microangiopathy. T2 and FLAIR detect 90% of acute infarcts by 24 h. MRA evaluates severity of artery stenosis, vascular occlusion, and collateral flow. DWI is the gold standard for evaluating ischemia. ADC quantifies the degree of diffusion to detect and assess infarct and distinguish new from chronic infarcts. PWI examines hemodynamic conditions at the microvascular level, enabling evaluation of penumbra. MR spectroscopy images intracellular metabolites and could serve as a surrogate for stroke treatment.	US evaluates vascular pathologies. Color duplex sonography provides hemodynamic information, such as stenosis, occlusion, and collaterals and morphological findings. Transcranial Doppler images major cerebral artery occlusions and monitors the effects of thrombolytic therapy. DSA evaluates vascular pathologies and therapeutically plays a vital role in catheterization procedures. SPECT and PET image physiology and may serve as a surrogate in differential diagnosis and hemodynamic assessment. SPECT lacks anatomy and has low resolution and relatively long acquisition time. PET is quantitative and is established as the gold standard to define infarct and penumbra; it is not employed in daily practice due to its high cost, lack of anatomy, long acquisition time, and limited availability.

## Data Availability

No new data were created.

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
