# Peer review of "Taxonomy of Acute Stroke: Imaging, Processing, and Treatment"

_diagnostics, 2024, doi:10.3390/diagnostics14101057_

Round 1

Reviewer 1 Report

Comments and Suggestions for Authors

I have the pleasure to review the manuscript "Taxonomy of acute stroke: imaging, processing, and treatment", submitted by Nowinski.

In my opinion, this manuscript does not meet the standards for publication and should be rejected.

I found it difficult to read and with several grammar mistakes. Moreover, it was unclear to me the main goal of the paper. It looks like an articulate list of diagnostic approaches to stroke without a real scientific purpose.

For all the above mentioned reasons it should be rejected.

Author Response

REVIEWER 1

I have the pleasure to review the manuscript "Taxonomy of acute stroke: imaging, processing, and treatment", submitted by Nowinski.

In my opinion, this manuscript does not meet the standards for publication and should be rejected.

I found it difficult to read and with several grammar mistakes. Moreover, it was unclear to me the main goal of the paper. It looks like an articulate list of diagnostic approaches to stroke without a real scientific purpose.

For all the above mentioned reasons it should be rejected.

Thank you for your frank opinion. I wish you could provide a more constructive report to support your decision. This is a review paper and not “an articulate list of diagnostic approaches to stroke without a real scientific purpose”. The goal of the paper was clearly stated in the Introduction. It is enhanced in the revised manuscript as follows:

The goal of this work is to capture the state-of-the-art in stroke management in three areas, namely, stroke imaging, image processing and analysis, and treatment. The methods in these areas are featured, widely illustrated, and categorized and their corresponding taxonomies are summarized in the form of comprehensive and straightforward diagrams. In addition, the introduced taxonomies are compared against other classifications.

Reviewer 2 Report

Comments and Suggestions for Authors

Dear author

Congratulations on the comprehensive review; it is very well written and explains the importance of imaging in managing stroke.

CT/MRI is used as a first-line diagnostic tool for stroke, and MR is used to observe the severity of arterial stenosis.

However, the article is well written. If possible, I will suggest adding a section for future development on the importance of portable imaging and its benefit on the primary healthcare system to diagnose or treat acute stroke cases.

Author Response

REVIEWER 2

Congratulations on the comprehensive review; it is very well written and explains the importance of imaging in managing stroke.

Thank you.

CT/MRI is used as a first-line diagnostic tool for stroke, and MR is used to observe the severity of arterial stenosis.

However, the article is well written. If possible, I will suggest adding a section for future development on the importance of portable imaging and its benefit on the primary healthcare system to diagnose or treat acute stroke cases.

Thank you for this important point. It is addressed in the revised manuscript as follows:

The diagnosis and treatment of acute stroke have changed dramatically during the last few years because of the recent advances in endovascular treatment (as reviewed by Widimsky et al. [139] in terms of key stroke trials, devices, and techniques) and portable stroke imaging which is crucial in time-critical situations. In particular, this includes the Mobile Interventional Stroke (MIST) Team model of stroke treatment that consists in traveling to perform endovascular thrombectomy in a hospital where the patient is first diagnosed. The MIST trial demonstrated that the use of a MIST was faster and led to improved discharge outcomes when compared with the drip-and-ship model i.e. transporting the patient to the nearest primary stroke center [140]. Another novel concept is a portable ultrasound tomography device for vascular brain lesions proposed by Kazmierski [141]. This device contains multiple transcranial ultrasound probes placed on the surface of the head allowing for a significant reduction of time from stroke onset to thrombolytic therapy.

Reviewer 3 Report

Comments and Suggestions for Authors

The present manuscript titled “Taxonomy of acute stroke: imaging, processing, and treatment” led by Wieslaw discussed different types of imaging technologies used to detect stroke in clinical settings. The present manuscript is well written and explains all the necessary information related to imaging technologies available to identify stroke. Though the manuscript addresses the current status of imaging technologies, the following modifications are to be made in the current version of the manuscript.

1.     Abstract

·       Line 7-8: The first line of the abstract is conclusive. Rewrite.

·       Line 8-9: Rewrite the sentence.

·       The abstract must discuss the importance of the present review and mention how this review will be useful to the scientific community.

2.     Line 40: Remove e.g.,

3.     Line 59-60: Merge 59-60 sentences with line 58.

4.     There are several two-line sentences without connections to previous sentences or forward sentences, for example, lines: 104-105, 117-118, 119-121, section 3.1.2.2, 306-307, 382-399, 472-473, 606-607, 623-624. Author needs to merge and rewrite the sentence.

5.     Add reference to section 3.1.2.

6.     Line 755-761: Authors need to give a conclusion at the end of the paragraph (The outcome of the data retrieval from Google Scholar, and PubMed must be mentioned).

7.     It would be advantageous for the current manuscript to add a table that describes the current imaging techniques and their shortcomings.

Comments on the Quality of English Language

Minor corrections.

Author Response

REVIEWER 3

The present manuscript titled “Taxonomy of acute stroke: imaging, processing, and treatment” led by Wieslaw discussed different types of imaging technologies used to detect stroke in clinical settings. The present manuscript is well written and explains all the necessary information related to imaging technologies available to identify stroke. Though the manuscript addresses the current status of imaging technologies, the following modifications are to be made in the current version of the manuscript.

  1. Abstract

Line 7-8: The first line of the abstract is conclusive. Rewrite.

Line 8-9: Rewrite the sentence.

The abstract must discuss the importance of the present review and mention how this review will be useful to the scientific community.

  1. Line 40: Remove e.g.,
  2. Line 59-60: Merge 59-60 sentences with line 58.
  3. There are several two-line sentences without connections to previous sentences or forward sentences, for example, lines: 104-105, 117-118, 119-121,

Done.

section 3.1.2.2, 306-307, 382-399, 472-473, 606-607,

Note that each of these texts addresses a different method. Consequently, each text forms a separate paragraph and is not connected to the previous sentence.

623-624. Author needs to merge and rewrite the sentence.

Done.

  1. Add reference to section 3.1.2.

The following references are added [34-62].

  1. Line 755-761: Authors need to give a conclusion at the end of the paragraph (The outcome of the data retrieval from Google Scholar, and PubMed must be mentioned).

The number of publications on PubMed was already given in the original manuscript. The number of AI acute stroke publications on Google Scholar of 1,830,000 is added.

  1. It would be advantageous for the current manuscript to add a table that describes the current imaging techniques and their shortcomings.

A table describing imaging techniques is added, Table 1.

Reviewer 4 Report

Comments and Suggestions for Authors

Comments: 

This is a review article that focuses on categorizing methods for stroke imaging and providing their taxonomies. The author has summarized all the imaging modalities in stroke evaluation. This review article is nicely written and covers all important aspects of the topic. The manuscript is well structured and easy to read. The author needs to address the following points: 

  1. The review mainly focusses on infarcts. However, the term stroke encompasses the hemorrhagic ones also. Hence necessary changes in the title and manuscript may be to mention that this review mainly focusses on imaging in infract. 

  1. Line 48-50, the sensitivity of CT and MRI should be mentioned in which type of stroke e.g ischemic or hemorrhagic. This would be clear to readers if such clarifications were made beforehand. 

  1. Appropriate references to the Taxonomy of stroke imaging processing should be provided. A brief overview of Non-atlas/Atlas image processing, their utility and differences should be provided before discussion of individual methods for better understanding of the readers. 

  1. The mechanism, usefulness, sensitivity and specificity of various methods may be provided in a tabular format for better readability. 

  1. It would be better if the use of CAD in emergencies and thrombolysis were discussed with relevant clinical trials or use in clinical settings. In case of no such data, the same may be mentioned, this would be more informative to the readers.

Comments on the Quality of English Language

Minor editing of English Language is required. 

Author Response

REVIEWER 4

This is a review article that focuses on categorizing methods for stroke imaging and providing their taxonomies. The author has summarized all the imaging modalities in stroke evaluation. This review article is nicely written and covers all important aspects of the topic. The manuscript is well structured and easy to read. The author needs to address the following points: 

  1. The review mainly focusses on infarcts. However, the term stroke encompasses the hemorrhagic ones also. Hence necessary changes in the title and manuscript may be to mention that this review mainly focusses on imaging in infract.

Hemorrhagic stroke is extensively addressed in the revised manuscript.

A new Section 3.1.2.3. is added addressing hematoma segmentation methods in hemorrhagic stroke.

3.1.2.3. Hematoma segmentation

The imaging properties of ischemic infarcts are different from those of hematomas and, consequently, several dedicated methods for hematoma segmentation in hemorrhagic stroke have been proposed. Hematomas on NCCT vary in shape, size, location, density (intensity), contrast, and texture. Therefore, segmentation of hematomas depends on many factors, including partial volume effect (volume averaging), fuzzy and low contrast borders, noise, beam hardening, motion artifacts, hematoma closeness to bone, head tilt, and incomplete head coverage. The characterization of the distribution of intraventricular and intracerebral hematomas in NCCT scans was studied by Nowinski et al. [57]. This study used 289 serial retrospective scans and provided the quantitative relationship of hematomas with respect to gray and white matters.

Bhanu Prakash et al. proposed a modified distance regularized level set evolution method for hemorrhage segmentation from NCCT [58]. The method performs preprocessing, including filtering and skull removal, and segmentation employing a distance regularized level set evolution with shrinking and expansion. The method was validated on 200 NCCT scans collected at 10 different hospitals within the Clot Lysis Evaluating Accelerated Resolution of intraventricular hemorrhage multicenter, randomized phase III clinical trial (CLEAR III). The scans were grouped as small, medium and large based on the volume of blood yielding the median DSI of 89.7% for the small group, 85.8% for the medium group, and 91.7% for the big group.

Patel et al. employed a 3-D convolutional neural network for the segmentation and quantification of intracerebral hemorrhage in NCCT using a combination of contextual information on multiple scales [59]. The method was evaluated on 2 datasets of 25 and 50 patients yielding a median DSC of 91% and 90%, respectively.

Bhanu Prakash et al. [60] proposed a texture-energy method for hemorrhage segmentation from NCCT. The method involves windowing, skull stripping, convolution with textural energy masks, and segmentation with combined thresholding and fuzzy C-means. The method validated on 201 NCCT scans (from the CLEAR III phase III clinical trial) yielded the median sensitivity, specificity, and DSC of 84.9%, 99.9%, and 87.1%, respectively.

An automatic segmentation approach based on machine learning employing a voxel-wise random forest classification was proposed by Scherer et al. [61]. Post-processing of prediction maps involved the identification of the predominant clot, removal of isolated islands/voxels by morphological operations, and Gaussian smoothing of the probability map boundaries. The algorithm agreement with the manual reference tested on 30 NCCT scans was strong with the concordance correlation coefficient of 0.95.

Bhanu Prakash et al. [62] compared three methods for intra-ventricular and intra-cerebral hemorrhage segmentation based on thresholding, clustering, and graph theory modified using textural energy-based normalization along with preprocessing (filtering, skull stripping) and postprocessing (artifact removal). The methods were tested on 201 NCCT scans (from the CLEAR III). The median sensitivity, specificity, and DSC were 86.19%, 99.94%, and 86.55% for the modified thresholding; 83.23%, 99.93%, and 84.10% for the modified fuzzy C-means; and 87.28%, 99.81%, and 79.17 % for the modified normalized cut method. The preprocessing and postprocessing enhanced the DSC by 10% and 3%, respectively. The use of textural energy along with the Hounsfield value in the modified methods increased the DSC by about 8-10%.

The following text is added regarding hemorrhagic stroke treatment:

There are also several therapies to treat hemorrhagic stroke especially since the mortality rate of acute intracerebral hemorrhage can reach up to 40% [122]. One of them is to evacuate the hemorrhage surgically by performing craniotomy. This procedure additionally facilitates decompression. A less invasive approach is to insert a catheter into the ventricular system and lyse the blood clot by administering t-PA [123] (see also Section 5.3).

In addition, a description of the hemorrhagic stroke CAD is added in Section 5.3:

The CAD system for hemorrhagic stroke aims at the progression and quantification of blood clot removal over time. It segments, quantifies, and displays hematoma in 2D and 3D, and supports evacuation of hemorrhage via thrombolytic treatment by monitoring progression and quantifying clot removal [133]. The procedure requires a catheter to be stereotactically inserted into the ventricular system, t-PA administered through it, and a series of NCCT scans acquired to monitor the outcome of clot lysis [123]. This CAD system supports 7-step workflow including select patient, add a new study, process patient’s scans, show segmentation results, plot hematoma volumes, show 3D synchronized time series hematomas along with their intraventricular and intracranial compartments, and generate report. The main components of the CAD system architecture are main application with the user interface, tools, and hematoma automated segmentation algorithms [58][60][62]. The tools include a contour editor to edit the segmented results, 3D surface modeler, 3D volume measure, histogramming, hematoma volume plot, and 3D synchronized time-series hematoma display and manipulation. The CAD system was employed in the CLEAR III and MISTIE III (Minimally Invasive Surgery plus rt-PA for Intracerebral Hemorrhage Evacuation) phase-III, multi-center clinical trials [123].

  1. Line 48-50, the sensitivity of CT and MRI should be mentioned in which type of stroke e.g ischemic or hemorrhagic. This would be clear to readers if such clarifications were made beforehand.

Done.

  1. Appropriate references to the Taxonomy of stroke imaging processing should be provided. A brief overview of Non-atlas/Atlas image processing, their utility and differences should be provided before discussion of individual methods for better understanding of the readers.

This overview is provided as follows:

Non-atlas/template-based methods are based on various principles and, consequently, are very heterogeneous. The simplest methods employ windowing for intensity and contrast transformations. The lesion segmentation methods utilize a variety of techniques such as thresholding, region growing, edge detection, clustering, textures, watersheds, and wavelets, among others. The AI-based techniques have been growing rapidly and include numerous approaches including linear regression, support vector machines, decision trees, random decision forests, k-nearest neighbors, k-means clustering, hidden Markov model, artificial neural networks, and convolutional neural networks with various architectures. Many of these methods focus on local changes caused by a stroke lesion. A different approach is taken in the global density-guided methods exploiting density sampling of the entire volume in numerous ranges. To increase the performance, many approaches are hybrid with some dominant technique. It shall also be noted that the imaging properties of ischemic infarcts and hematomas are different, so the handing of these two types of stroke lesions requires different approaches.

The practical usefulness of these methods depends on their performance. Unfortunately, many of them briefly outlined below are not validated (especially on diverse and multi-centered data) at all or the validation is on small samples. Even if validated, the validation is on various datasets and often with different performance measures. Consequently, the complete assessment and comparison of the presented below methods are difficult if possible at all.

  1. The mechanism, usefulness, sensitivity and specificity of various methods may be provided in a tabular format for better readability.

The general mechanisms of methods are simply and clearly captured in the taxonomy diagrams, so there is no need to introduce redundancy.

The idea of providing sensitivity and specificity of various methods in a tabular format would be good provided that the authors consistently report these parameters in their works. But this is not the case. Most papers do not provide validation of their methods at all. Second, even if validated, methods are validated on different datasets in terms of number and modality which makes their comparison impossible. Third, methods use different measures, not only sensitivity and specificity but also other measures, such as accuracy and DSC; moreover, these measures can quantify various features. Consequently, any attempt to provide this “better readability” would result in the elimination of the majority of works and the comparison of the remaining could potentially be confusing.

  1. It would be better if the use of CAD in emergencies and thrombolysis were discussed with relevant clinical trials or use in clinical settings. In case of no such data, the same may be mentioned, this would be more informative to the readers.

The manuscript is reorganized and the CAD system description is moved from Section 3.2.2.3. to new Section 6. In addition, this section is extended and also a CAD system for hemorrhagic stroke is featured.

Round 2

Reviewer 1 Report

Comments and Suggestions for Authors

I have previously rejected this paper